# ε^2^-Phages Are Naturally Bred and Have a Vastly Improved Host Range in *Staphylococcus aureus* over Wild Type Phages

**DOI:** 10.3390/ph14040325

**Published:** 2021-04-02

**Authors:** David Sáez Moreno, Zehra Visram, Michele Mutti, Marcela Restrepo-Córdoba, Susana Hartmann, Ana Isabel Kremers, Lenka Tišáková, Susanne Schertler, Johannes Wittmann, Benham Kalali, Stefan Monecke, Ralf Ehricht, Grégory Resch, Lorenzo Corsini

**Affiliations:** 1PhagoMed Biopharma GmbH, Leberstrasse 20, A-1110 Vienna, Austria; david.saez@phagomed.com (D.S.M.); Zehra.visram@phagomed.com (Z.V.); Michele.mutti@phagomed.com (M.M.); Marcela.restrepo@phagomed.com (M.R.-C.); susana.hartmann@phagomed.com (S.H.); kremers.ana@gmail.com (A.I.K.); Lenka.tisakova@phagomed.com (L.T.); 2Leibniz Institute DSMZ—German Collection of Microorganisms and Cell Cultures, Inhoffenstraße 7B, 38124 Braunschweig, Germany; susanne_schertler@web.de (S.S.); jow12@dsmz.de (J.W.); 3Bactrace Biotec AG, Neherstr. 1, 81675 Munich, Germany; b.kalali@bactrace.com; 4Leibniz Institute of Photonic Technology (IPHT), 07745 Jena, Germany; stefan.monecke@leibniz-ipht.de (S.M.); Ralf.Ehricht@leibniz-ipht.de (R.E.); 5Institute of Medical Microbiologye and Hygiene, Faculty of Medicine Carl Gustav Carus, Technical University Dresden, Fiedlerstr. 42, D-01307 Dresden, Germany; 6InfectoGnostics Research Campus Jena, 07743 Jena, Germany; 7Institute of Physical Chemistry, Friedrich-Schiller University, 07743 Jena, Germany; 8Department of Fundamental Microbiology, University Lausanne, CH-1015 Lausanne, Switzerland; gregory.resch@unil.ch

**Keywords:** phage therapy, phage breeding, phage training, homologous recombination, host range, antimicrobial resistance, phage cocktail, *S. aureus*, MRSA, MSSA

## Abstract

Due to the rapid spread of antibiotic resistance, and the difficulties of treating biofilm-associated infections, alternative treatments for *S. aureus* infections are urgently needed. We tested the lytic activity of several wild type phages against a panel of 110 *S. aureus* strains (MRSA/MSSA) composed to reflect the prevalence of *S. aureus* clonal complexes in human infections. The plaquing host ranges (PHR) of the wild type phages were in the range of 51% to 60%. We also measured what we called the kinetic host range (KHR), i.e., the percentage of strains for which growth in suspension was suppressed for 24 h. The KHR of the wild type phages ranged from 2% to 49%, substantially lower than the PHRs. To improve the KHR and other key pharmaceutical properties, we bred the phages by mixing and propagating cocktails on a subset of *S. aureus* strains. These bred phages, which we termed evolution-squared (ε^2^) phages, have broader KHRs up to 64% and increased virulence compared to the ancestors. The ε^2^-phages with the broadest KHR have genomes intercrossed from up to three different ancestors. We composed a cocktail of three ε^2^-phages with an overall KHR of 92% and PHR of 96% on 110 *S. aureus* strains and called it PM-399. PM-399 has a lower propensity to resistance formation than the standard of care antibiotics vancomycin, rifampicin, or their combination, and no resistance was observed in laboratory settings (detection limit: 1 cell in 10^11^). In summary, ε^2^-phages and, in particular PM-399, are promising candidates for an alternative treatment of *S. aureus* infections.

## 1. Introduction

Bacteriophages have been proposed as antimicrobial therapeutics for a long time [1,2], because of their efficacy against antimicrobial resistant strains, their effectiveness on biofilms [3], their safety and their microbiome-sparing specificity [4]. To be effective, therapeutic phages must not only be able to propagate on the bacterial strain of interest, but their virulence also needs to be sufficient to quantitatively reduce the target bacterial population in vivo [5,6], and to avoid pathogenic, bacteriophage-insensitive mutants (BIM) overgrowing the original strain in situ [7,8].

The limited host range and resistance formation of wild type phages is a technical complication that can lead to the need for cocktails [9,10]. However, combining 14 phages into one cocktail made it difficult to control their stability, which contributed to the failure of a clinical trial [11]. In the personalized approach to phage therapy, where patients are treated either with single phages or with personalized cocktails [12], a broad host range and low resistance formation would reduce its operating cost, and may even increase the efficacy of the approach [13,14].

Multiple approaches have been proposed to improve the properties of wild type phages, including phage engineering [15,16] and directed evolution of phages (“phage training”) [14,17,18]. Phages can be rationally engineered to increase the host range by promoting the adsorption step, for example by mutating host-range determining regions in the tail spike protein [19]. In a different approach, a CRISPR-Cas3 system was introduced into *Clostridioides difficile* phages, which increased their virulence by reducing the rate of abortive infections and lysogeny [20]. In another application, lysogenic phages were modified to obtain obligately non-temperate phages which were successfully used to treat a disseminated *Mycobacterium abscessus* infection [21].

While the rational engineering of phages is promising, it is limited by the understanding of the phage genomes, as the function of ~50–80% of genes on a phage is often unknown [22,23]. Therefore, strategies to improve the characteristics of wild type phages based on random changes of the genome have been proposed. In one study, phages with improved heat stability were generated by random mutagenesis followed by a selection filter [24]. Moreover “phage training” by sequential re-propagation, and selection for advantageous mutations has been described [14]. This strategy was adopted to improve the host range of *S. aureus* phage 812 [25]. A different approach is based on random intercrossing of whole genomic regions of phages. It can be mediated by homologous recombination inside bacterial cells, upon superinfection with multiple phages. This method was used more than 70 years ago to “cross-breed” *E. coli* phages with different plaque morphologies [26]. In nature, this process leads to the well-described mosaicism of genomes of phages of all classes [27]. Promoting recombination by superinfection, combined with an appropriate selection filter, was also used to increase the host range of phages infecting *P. aeruginosa*, by what is called the Appelman’s protocol [17,18].

*S. aureus* was selected by the WHO as a pathogen against which alternatives to antibiotics should be developed with high priority [28]. It is a frequently antibiotic-resistant and biofilm-forming bacterium, which is particularly problematic when infecting implants [29], in systemic infections [30] or in atopic dermatitis [31] and ulcers [32]. *S. aureus* is a clonal organism, meaning that traits are passed on more frequently by vertical gene transfer than horizontal gene transfer [33]. Therefore, different strains are well characterized by their clustering to clonal complexes (CC) and sequence types (ST) [34]. The epidemiology of CCs in humans is distinct from livestock [35]; 25% of humans are asymptomatic carriers, and few differences have been found in the distribution of CCs between commensal strains on the skin microbiome and infections [34,36,37].

*S.aureus* phages belonging to the *Herelleviridae* family have been analyzed in several studies. Phage 812 [25,38], ISP [39] (closest identity to 2002, used in our study) and phiIPLA-RODI [40] (closest identity to phage BT3, used in our study) are well-characterized lytic phages with *Myoviridae* morphology. Remus and Romulus [41], two phages that are very similar to each other but genetically unrelated to the other known *Herelleviridae*, also have potential for use as antimicrobials.

Moreover, many studies have reported *S. aureus* phages with a broad plaquing host range in multiple panels of strains [22,25,39,41,42,43,44]. While in our study, the broad plaquing host range could be reproduced, the wild type phages could prevent bacterial growth in suspension (which we called the kinetic host range, KHR) only for 2–49% of 110 *S. aureus* strains tested. Therefore, we improved these phages for therapeutic use, by intercrossing different wild type phages by superinfection, and by selecting the best progeny for the improved host range, increased virulence and reduced resistance formation. We named these naturally bred, non-wild type phages ε^2^ (evolution squared)-phages and showed that their virulence and KHR are strongly improved. The cocktail PM-399, composed of three ε^2^-phages, clears 92% of 110 *S. aureus* strains in suspension over 24 h. In summary, ε^2^-phages, and in particular PM-399, are promising candidates for an alternative treatment of *S. aureus* infections.

## 2. Results

### 2.1. Study Conducted on a Panel of S. aureus Strains Which Reflects the Clonal Complex Prevalance in Human Infections

To characterize and optimize the host ranges of phages intended for treatment of human infections, we compiled a panel of 110 *S. aureus* strains, which reflects the global epidemiology of CCs causing human infections as closely as possible. This was done by considering the distribution of CCs found in epidemiological studies conducted in various regions [37,45,46,47]. The composition of the panel used for the experiments described in this study is depicted in Table 1 and Appendix A.

### 2.2. The Plaquing Host Range of Wild Type Phages Is Much Higher Than Their Kinetic Host Range

We compiled a panel of *Herelleviridae* wild type phages isolated in our labs and from public collections. These included the previously described phages 812 [38], Remus, and Romulus [41], as well as the previously not described phages vB_HSa_2002 (from here onwards 2002) and BT3. We also included the two *Podoviridae* phages P66 and P68 [22,48]. The efficiency of plaquing (EOP) of each of the seven phages was measured against our panel of *S. aureus* strains. The plaquing host range (PHR) was defined as the percentage of strains where plaques were formed at any EOP, while strains where no plaques formed but the phage spot locally impaired growth (“opaque lysis”) were not counted for the PHR. The PHR of the five *Herelleviridae* wild type phages on the panel of *S. aureus* ranges between 51% (Romulus) and 60% (BT3, 2002, Remus), as depicted in Figure 1A, with more detailed results for each strain depicted in Appendix A.

We also tested to what extent phages could inhibit the growth of bacteria in suspension, by measuring what we termed the kinetic host range (KHR). The KHR was defined as the percentage of strains for which suspensions were cleared after 24 h of incubation at 37 °C. As depicted in Figure 1A, the KHRs of the wild type phages were remarkably lower than the PHRs, ranging between 2% (phage BT3) and 49% (Romulus). Interestingly, phage Romulus had both the lowest PHR and the highest KHR among the tested phages. The *Podoviridae* P66 and P68 had PHR values comparable to the *Herelleviridae*, but both had a KHR of 0% (Appendix A), indicating bacterial growth in the presence of these phages.

Next, we analyzed the correlation between EOP and the ability to inhibit the growth of bacteria in suspension on each of the 110 *S. aureus* strains of the panel. As shown in Figure 1B–D, the correlation between the EOP and OD ratio (defined as the ratio of turbidity of treated vs. untreated suspensions after 24 h), is weak. As expected, if a phage does not form plaques or only “opaque lysis” is formed, the OD ratio is close to 1, indicating no or just a small difference between phage-treated and non-treated suspensions. But in cases of measurable EOP, there is no correlation with the OD ratio, indicating that on many strains, the phages propagate well (EOP close to 1), but, at least under the conditions of the experiment, fail to substantially reduce the bacterial population over the course of 24 h.

### 2.3. Phage Breeding Strongly Increased the Kinetic and Plaquing Host Range

Since the wild type phages, despite forming plaques, were not able to suppress bacterial growth in suspension on many strains, we improved the KHR by breeding the phages. The phages 812, 2002 and BT3 (three *Kayvirus*) were mixed and bred as a cocktail, while phages Remus and Romulus, belonging to the *Silviavirus* genus, were not expected to recombine with *Kayvirus*, and thus were bred individually. After 20 consecutive rounds and isolation by single plaque re-streaking, the resulting bred phages (termed evolution squared (ε^2^) phages) were propagated on a common host strain. The *Podoviridae* P66 and P68 were bred individually and as a cocktail.

Out of the 56 phages isolated from the cocktails of *Herelleviridae*, the 10 ε^2^-phages with broadest KHR values were analyzed further (8 progeny phages from the mix of 2002, 812 and BT3, 1 progeny from Remus, 1 from Romulus, Table 2). The ε^2^-phages derived from the mix of 2002, 812 and BT3 had a KHR of up to 64% (phage PM4, Figure 2). This is an improvement of 99% compared to the KHR of the most active ancestor, 2002, of 32%. Phages PM56 and PM93, derived from Romulus and Remus, respectively, had KHR values of 62% (26% improvement over Romulus) and 65% (75% improvement over Remus), respectively (Figure 2). Moreover, the PHR increased for the most promising ε^2^-phages compared to the ancestors, with values up to 81% for PM4 (Appendix A), improved by 35% over the best ancestors 2002 and BT3. A more detailed view on KHR and PHR of all phages can be found in Appendix A.

Interestingly, the KHR generally improved both on strains used for breeding, but also on strains not used for breeding, which the ε^2^-phages had never encountered (Figure 2).

This indicates that the breeding process substantially increased the ability of the phages to control/reduce bacterial populations in suspension, potentially increasing the virulence of the phages and/or reducing the ability of the bacteria to become phage tolerant.

In contrast, the *Podoviridae*, bred individually or as a cocktail, did not show KHR improvements and never suppressed growth of any strain over 24 h (data not shown). Therefore, bred *Podoviridae* phages were not isolated and excluded from further analyses. The rate of spontaneous resistance measured for the ancestor P66 was >1 cell in 10^4^, in line with the observation of a KHR of 0%.

### 2.4. ε^2^ Phages Have Increased Virulence Compared to Wild Types in Bacterial Suspensions

The increased KHR of the ε^2^-phages indicates that, at a multiplicity of infection (MOI: number of phages per bacterial cell) of 10, these phages inhibited the growth of more *S. aureus* strains than the ancestor phages (Figure 2). The strains that were not inhibited over 24 h were either not susceptible or BIMs were formed over the course of the infection. Therefore, we dissected these effects for a subset of three strains for which growth in suspension was suppressed by ε^2^-phages but not by ancestors.

For three of these strains, we measured phage virulence curves and the virulence index as described [5]. We used PM4, one of the most promising ε^2^-phages, and its ancestors 2002, 812, and BT3. Virulence was analyzed at MOIs ranging from 1 to 10^5^ as described in Figure 3A. As shown in Figure 3B, phage PM4 fully suppresses the growth of strain CC239-MRSA-III (2017-046) at a MOI as low as 10 for 24 h, while the ancestors 2002 and 812 each need a MOI of 10,000 to achieve the same effect. BT3 cannot suppress the growth of this strain at any MOI. Figure 3C depicts the local virulence curve calculated from the data of experiments of Figure 3B and shows that the virulence of PM4 is higher than the virulence of any of its ancestors at any MOI for this strain. Analogous measurements were done for strains CC30-MRSA-IV (2011-278) and CC25-MSSA (B91) in Figure 3D–G, respectively, with similar results: the virulence of PM4 is strongly increased on all these strains in comparison to its ancestors. The increased KHR of PM4 vs. its ancestors (see Figure 2), on the full panel of 110 strains at MOI = 10, indicates that the virulence increased on more than the three strains analyzed in detail in Figure 3. Absence of bacterial growth over 24 h for 64% of the strains when treated with PM4 also indicates that no (fast-growing) BIMs could form on any of these strains.

We noticed that over the course of the infection, outgrowth of the bacteria was occurring (e.g., phage 2002 on strain CC25-MSSA (B91) at MOI 100 in Figure 3F). However, phage susceptibility tests after re-streaking of the outgrowth could not identify bacteriophage-insensitive mutants (BIMs) against 2002, 812 or PM4 (data not shown). The reason why these phage-susceptible bacteria were not eradicated, despite a very high final MOI, is elusive. It might hint at the acquisition of a transient tolerance against the phages, such as the phenomenon recently described for *B. subtilis* [49].

### 2.5. The Cocktail PM-399 Has a KHR of 92% and Does Not Form Resistance at a Lab Scale

Because PM4, PM56 and PM93 are the most complementary combination of phages by KHR, we combined them into a cocktail, which we called PM-399. PM-399 has a KHR of 92% of the panel of 110 *S. aureus* strains (Figure 4A).

We also compared the propensity of resistance formation by passaging of *S. aureus* ATCC43300 (CC30/ST39-MRSA-II) against PM-399 and the antibiotics vancomycin and rifampicin, which are routinely used against MRSA implant associated infections. While Rifampicin led to full resistance formation (MIC > 512 μg/mL) already after 2 rounds of passaging, the combination of Rifampicin and Vancomycin allowed an increase in MIC by a factor of 8 over 11–20 rounds (from an initial MIC of 1 + 0.016 μg/mL to 8 + 0.128 μg/mL of Rifampicin and Vancomycin, respectively). No increase in MIC was observed for PM-399 (Figure 4B).

The rate of spontaneous resistance formation of *S. aureus* ATCC43300 (CC30/ST39-MRSA-II) was tested against PM4, PM93 and the cocktail PM-399, as depicted in Appendix A. Bacterial suspensions of 10^11^ CFU (1 L at OD_600_ 0.2) were infected at a MOI of 10 and incubated for 24 h, in triplicate, and the suspensions were always clear at the end of the experiment. Single bacterial colonies surviving phage infection were tested for phage susceptibility by spotting. All colonies tested (10 for each phage/cocktail) were still susceptible to the phages, indicating that no BIMs formed against any of the phages in PM-399 (limit of detection of 10^11^ CFU).

### 2.6. Wild-Type and Bred ε^2^-Phages Phages Belong to the Herelleviridae Family and Are Deemed Safe for Therapeutic Use

Wild-type *Herelleviridae* and ε^2^-phages were sequenced by OxNanopore and Illumina (Table 2). All phages analyzed in this study have long-terminal repeats of 5–9 kbp at both ends of the genomes. Phage 2002 was isolated in Lausanne, Switzerland, and was not described previously. It has a genome length of 145,076 bp and an identity of 99% to phage ISP (FR852584.1). Phage BT3 is a new phage isolated in Braunschweig, Germany, and has a genome length of 141,567 bp. Its closest published homolog is phage philIPLA-RODI [40] with 96.93% identity on the nucleotide level. Both BT3 and 2002 were classified as members of *Kayvirus* in the *Herelleviridae* family by sequence similarity.

Apart from a lytic life cycle, therapeutic phages should be free from genes encoding virulence factors, antimicrobial resistance or toxins [50]. All *Herelleviridae* phages infecting *S. aureus* have already been described as strictly non-temperate and safe for therapy [51]. To confirm this, we analyzed the phage life cycle with PHACTS, screened the genome for antibiotic resistance genes with CARD and ResFinder 4.1, and for virulence factors in Virulence Finder 2.0. These analyses confirmed that the newly described ancestors 2002 and BT3 as well as all ε^2^-phages are suitable for phage therapy.

### 2.7. ε^2^-Phages with the Best Host Ranges Have Genomes Intercrossed from up to Three Ancestors

Genomic analysis indicates that the ε^2^-phages which are progeny from the 2002/812/BT3-mix and have the broadest kinetic host range, have a genome intercrossed from up to three different ancestors. As depicted in Table 3, the genomes of the ε^2^-phages can be subdivided into individual stretches of lengths between 585 and 30,589 bp (based on PM4 sequence), where most stretches are 100% identical to one of the three ancestor phages. Each of the 7 ε^2^-phages depicted in Table 3 have a different mosaicism, meaning that different genome stretches are inherited from different ancestor phages. Some stretches have several mutations, e.g., stretch 7 in PM4 is 99,4% identical to 2002, 5 bp are mutated over 834 bp (not shown).

Figure 5A shows a more graphical view of stretches 24, 25 and 1 to 7 for phage PM4. In this genomic region of ~19 kb, there are at least six recombination sites, where the PM4 genome switches from one ancestor to another (2002 to BT3, back to 2002, then back to BT3, back to 2002, then to 812 and eventually back to 2002).

Stretches 1–7 correspond to the LTR (long terminal repeats) and are repeated at both ends of the genome. Interestingly, the recombination sites between the ancestor phages seem to be clustered in the LTRs. For example, PM4 has four recombination sites in the 8343 bp of the LTR (48 recombination sites per 100,000 bp), while it has only five recombination sites in the 131,941 bp between the LTR (4 recombination sites per 100,000 bp). The same is true for the other 6 ε^2^-phages, which have an average of 54 recombination sites per 100,000 bp in the LTR but only 6 per 100,000 bp in the genome part between the LTRs.

Figure 5B shows the genomic region from 56 to 87 kbp for phage PM4 (corresponds to stretches 19 to 21 in Table 3), encoding for tail-associated and receptor binding proteins (RBPs). Other genes encoding for putative major tail proteins were found in other locations (Locus tag 035, Locus tag 160–162). No recombination events nor mutations were found in any of these regions, all nucleotides were identical to the ancestor 2002.

The PM4 lysis module mainly comes from ancestor 2002, with one recombination event where genome switches to BT3, and then back to 2002, as shown in Figure 5C. This module comprises genes encoding for a transglycosylase (Locus Tag 071), a putative HNH endonuclease (Locus Tag 072), two putative membrane proteins (Locus Tag 073 and 074), a gene with unknown function (Locus Tag 075), the endolysin (locus Tag 076 and 078) interrupted by an intron (encoding an endonuclease [52], Locus Tag 077) and the holin (Locus Tag 079). We assume that, as described for other phages [41,52], the two ORFs encoding the endolysin are merged upon splicing of the mRNA.

The endolysin of PM4 could come from either 2002 or 812. Interestingly, the BT3 endolysin-coding gene does not have the intron. However, the last codon of the PM4 endolysin is an insertion from BT3 (not present in either 2002 nor 812). This adds the amino acid Glycine to the C-terminus of the endolysin (SH3b cell-wall binding domain).

We found a single point mutation in the transglycosylase gene (base G32,244A) translating to an amino acid change (T14I), not present in any of the ancestors.

Another gene encoding for a lytic enzyme (transglycosylase) was identified outside of the lytic module (Locus Tag 043) and was 100% identical to the ones in 2002 and 812.

### 2.8. Single Point Mutations on ε^2^-Phages Bred from Romulus and Remus Increased the Host Range

To attempt a different breeding process, Remus and Romulus were not bred as a cocktail, but individually. Their progeny ε^2^-(PM56 and PM93, from Romulus and Remus, respectively) do not exhibit the level of recombination identified for the progeny of the cocktail of 2002/812/BT3. The KHR of the ε^2^-phages PM56 and PM93 increased from 37% (Remus) to 65% (PM93), and 49% (Romulus) to 62% (PM56), as described in Figure 2 and Appendix A. These increases can therefore be attributed to single point mutations or short insertions/deletions described in Appendix A. Interestingly, two genes are mutated at the same site in both PM56 (progeny of Romulus) and PM93 (progeny of Remus), where D302 of the locus tag 046 on both Remus and Romulus were mutated to Y in both progeny phages. Locus tag 046 is located in the structural module of the genome, immediately after the cluster of genes predicted to encode for tail proteins, it was annotated as “structural protein” in Remus (JX846612), and it has 99% and 98% identity to APC42937.1 and BBI90234.1, respectively, which are both annotated as “capsid and scaffold protein”. Another common mutation was found in the locus tag 117 in Remus (G161V) and in Romulus (M180I), both genes encoding for the pentapeptide repeat-containing protein. In addition, PM56 has mutations conferring amino acid changes in a major capsid protein (F25S in the locus tag 018), in two hypothetical proteins (N129K in the locus tag 161 and an isoleucine insertion after I16 in the locus tag 187), and in an endonuclease (asparagine insertion after Y241 in the locus tag 081). PM93 has additional mutations conferring amino acid changes in the gene encoding for a hypothetical protein (V44L in the locus tag 062).

## 3. Discussion

*S. aureus* is responsible for life-threatening infections, and phages suited for therapy hold great potential as an alternative treatment. In this study, we describe ε^2^-phages; bred *S. aureus* phages with virulence and KHR vastly improved compared to their wild type ancestors. The three-phage cocktail PM-399 has a kinetic host range of 92%, measured on a panel of 110 *S. aureus* strains. The panel was composed to represent the natural diversity in human epidemiology. Local and regional differences in the CC distribution have been described [53], which are however likely to be a snapshot of a distribution that may vary over time. We believe that the panel is also representative for phage susceptibility, because susceptibility should be independent of the region where strains of a given CC were collected from.

A first surprising finding of this study is the lack of correlation between EOP and killing of planktonic bacteria (Figure 1). Even if plaques are formed on a given strain, in suspension, the bacteria may grow faster than the phages lyse them even at a starting MOI of 10. Currently, the suitability of individual phages for human phage therapy is mostly determined solely based on plaquing assays (“phagogram”) [54,55,56,57,58]. However, if the ability to form plaques does not indicate that the phage is able to control growth or even reduce the bacterial population, a phagogram by plaquing may not be the appropriate sole predictor for therapeutic success. This has been discussed previously [59], and some authors have already used kinetic methods to complement plaquing in the screening for suitable therapeutic phages [60].

Our analyses show that the rate of spontaneous resistance of P66 is >1 cell in 10^4^. This high rate of resistance formation may apply also to other *S. aureus Podoviridae*, as the KHR of all *Podoviridae* we tested was 0%, indicating that none of these phages could avoid resistant outgrowth over 24 h. Resistance formation by *S. aureus* against *Podoviridae* was shown to be mediated by changes in the expression level of enzymes glycosylating wall teichoic acids [61], and there is no hint that these BIMs might be less virulent or antibiotic resistant than the wild types. The inability of *S. aureus Podoviridae* to control growth of *S. aureus* over 24 h may pose a challenge to phage therapy with this family of phages, which is why we excluded them from our cocktail.

This study describes for the first time the phage breeding by the principles of the Appleman’s protocol [17] applied to phages infecting *S. aureus*. As an improvement to previous reports, a bacterial panel with a defined diversity was used to measure and select for broader host ranges. Starting the breeding process with wild type phages that were known to have high sequence similarity (812, 2002, BT3) increased the likelihood of homologous recombination. To our knowledge, this is also the first study demonstrating an increase in virulence, PHR and KHR upon phage breeding. The virulence of a phage is dependent on environmental conditions such as temperature and medium [5], but, keeping these factors constant, we could demonstrate that breeding improved the virulence of ε^2^-phage PM4 vs. its ancestors. The increased virulence of PM4 hints at a more efficient replication cycle, were either of adsorption rate, latency period and/or burst size are improved. The improved KHR of the ε^2^-phages may also hint at a lower rate of resistance formation compared to the ancestors. We found that for the ε^2^-phages PM4, PM93, and the cocktail PM-399, the rate of spontaneous resistance formation was lower than the limit of detection of 1 cell in 10^11^. In a human infection, it is very unlikely to reach a bacterial burden of 10^11^ cells [62] so that resistance formation against PM-399 in vivo is improbable. Further analysis is needed to investigate whether or not co-evolution between phage and bacteria follow different trajectories in vivo [63].

In contrast to the report by Burrowes et al. [17], we show that the KHR of ε^2^-phages, improved not only on the strains used for breeding, but also on the other strains, which the phages had never been in contact with. Therefore, we conclude that the improved pharmacological properties of the ε^2^-phages are valid for the *S. aureus* strains most relevant in human infections.

The analysis of the genomes revealed that the ε^2^-phages bred from the wild types 2002, BT3 and 812 are mosaics of their ancestors, with up to 20 recombination sites across the genomes. Given the low share of phage genes for which a function is known, it is impossible to explain why the ε^2^-phages have a higher KHR and virulence than their ancestors based on these data. This also means that it would have been impossible to predict the genetic changes required to improve the ancestor phages. In this sense, the breeding method described here, which is based on random changes and evolutionary selection, has an advantage over more rational approaches to phage engineering [15].

Previous studies have recognized phage tail fiber proteins and RBPs as key elements for host specificity [64,65,66]. However, we could not find any mutation in the ε^2^-phages in these genes. Interestingly, some recombination events were identified in the lysis module, and we observed that the LTRs were the regions carrying most of the mutations. LTRs encode for “host takeover” genes [67], essential for the infection process. This supports the idea that also the LTRs influence the host range expansion, as shown for individual mutations for phage 812 [25,38]. However, we suspect that the expansion of the host range shown in this study is driven by the accumulation and interaction of multiple recombinations and mutations rather than by single events. The single phage breeding (Romulus and Remus) also led to an increase in the host range. A similar host range expansion by multiple point mutations was described for *S. aureus* phages Sb1, attributed to a hypervariable complex repeat structure in the genome [68]; for phage 812, different host range-mutants showed mutations in the LTRs and in tail proteins [25,38]; and for phages infecting *Pseudomonas aeruginosa* and *Escherichia coli*, the host range expansion was based on tail fiber mutations [19,64]. Previous studies have hypothesized *S. aureus* resistance to phages to be caused mainly by restriction-modification systems, and less frequently by adsorption inhibition [38,69], which could be a possible reason for the lack of mutations in tail proteins found in this study.

In conclusion, this study describes ε^2^-phages lysing *S. aureus*, bred from wild type ancestors, and selected for a superior killing in suspension (KHR) and higher virulence than the wild type phages. It also describes the cocktail PM-399, composed of particularly potent ε^2^-phages with a KHR of 92%, a PHR of 96% and a resistance rate below 1 cell in 10^11^. This is the first time that breeding by homologous recombination is described for *S. aureus* phages and shows that it is possible to substantially improve key phage properties required for therapeutic success, by applying directed evolution, with limited knowledge of the underlying functions of the altered parts of the genomes. ε^2^-phages may not only increase therapeutic efficacy via the increased virulence, but, due to the increased kinetic host range, may also reduce the number of phages required in a fixed cocktail or in a phage bank for personalized phage therapy. Overall, the ε^2^-phages and the cocktail PM-399 are promising candidates for an alternative treatment of *S. aureus* infections.

## 4. Materials and Methods

### 4.1. Collection of S. aureus Strains and Growth Conditions

*S. aureus* strains used in this study are summarized in Table 1. *S. aureus* strains were grown in Brain Hearth Infusion (BHI, Carl Roth, Graz, Austria) broth or agar, with aeration, at 37 °C. Growth medium was supplemented with either vancomycin (Pfizer, Vienna, Austria) or rifampicin (Carl Roth, Graz, Austria) for MIC determinations.

### 4.2. Collection of Phages

812, Remus, Romulus and P66 were obtained from the d’Herelle collection at the Laval University in Canada. P68 was obtained from NCTC (Public Health England, London, UK). Phage 2002 was isolated in Lausanne, Switzerland from hospital waste water as described in [70]. Phage BT3 was isolated at the Leibniz Institute DSMZ—German Collection of Microorganisms and Cell Cultures (Braunschweig, Germany) in 2017, and also from hospital waste water. All phages used in this study are described in Table 2.

### 4.3. Propagation of Phages and Cocktail Preparation

All phages were propagated on their respective *S. aureus* host strain in suspension, by infecting a bacterial suspension at OD 0.01. After incubation at 200 rpm and 37 °C overnight or until clearance of the culture, the lysate was centrifuged at 10,000 rpm for 20 min, filtered with a 0.22-µm membrane filter, and stored at 4 °C.

The cocktail PM-399 was prepared by mixing PM4, PM56 and PM93 (1:1:1) to achieve the final total concentration as described in each assay, so that the concentration of each constituent phage is one third of the total.

### 4.4. Plaquing Host Range and Efficiency of Plaquing

Plaquing host range was assessed by spotting a 2 µL drop of phage suspension in serial dilutions with 0.85% NaCl from 1 × 10^7^ PFU/mL to 1 × 10^1^ PFU/mL onto a pre-seeded lawn of log phase bacterial cells of the strain under investigation in 0.5% top agar. After overnight incubation at 37 °C, plaques were counted at the appropriate dilution. EOP was calculated as the ratio of the PFU measured on the strain under investigation and the host strain used for propagation. The plaquing host range (PHR) was defined as the percentage of strains where plaques were formed at any EOP, where strains with no plaque formation but only impairment of growth at the site of the phage spot (“opaque lysis”) were not counted as plaque-forming. The total of strains tested was 110. Phage P68 was tested in a subset of 63 strains.

### 4.5. Kinetic Host Range

*S. aureus* strains were grown to a log phase. The suspensions of planktonic bacteria were diluted to 5 × 10^6^ CFU/mL, of which 90 µL were transferred to a 96-well or 384 well microplate. 10 µL phage solution adjusted to 5 × 10^8^ PFU/mL (total phage concentration, i.e., a third of the value for each phage in a cocktail) was added. The final reaction volume was 100 µL, the final concentration of bacteria was OD600 = 0.01 (corresponds to 5 × 10^6^ CFU/mL) in BHI medium (MOI = 10).

Reaction plates were incubated at 37 °C for 24 h. OD600 was measured with a TECAN microplate reader and further analyzed in Excel and Tableau. The KHR was calculated as the percentage of strains for which after 24 h, the OD600 of the phage treated sample was less than 10 percentage of the OD of the untreated bacterial growth control, where the KHR was measured on 110 *S. aureus* strains in duplicate, and the average of the two KHR values is reported in the figures.

All phages were tested in the panel of 110 strains, except *Podoviridae* phages P66 and P68, which were tested in a subset of 24 out of the 110 strains.

Statistical analysis was done assuming a Gaussian distribution of the KHR and PHR data (D’Agostino-Pearson normality test). PHR and KHR data were compared between ancestors and bred phages, as well as values of PHR and KHR of each phage, by one-way ANOVA in GraphPad Prism 9. Differences were considered significant if the *p* value was below 0.05.

### 4.6. Breeding

Breeding was conducted essentially as described in [17,18]. Briefly, one or more ancestor bacteriophages were pooled to create the input phage mixture, which was diluted or undiluted and mixed with each bacterial strain of the subpanel of 17–24 *S. aureus* strains in a 96-well microtiter plate. 100 µL of bacterial culture of a single strain in BHI at OD600 = 0.2 was mixed with 100 µL of a single phages or cocktail in serial 10-fold dilutions (undiluted to 10–3, starting at 1 × 10^9^ PFU/mL).

After incubation at 37 °C for 24 h, clear lysates and the first dilution of turbid lysates were pooled and used to infect the next round of breeding. The process was repeated 20–30 times. If no improvement, i.e., no clear wells on previously unsusceptible bacterial strains, was visible after 4–6 rounds, the bacterial strains were exchanged.

Single individual phages were isolated from the bred mixed lysate by re-streaking in host strain at least 5 times.

### 4.7. Resistance Formation

The cocktail PM-399 was prepared by combining three bred bacteriophages PM4, PM56 and PM93 at 1:1:1 ratio. *S. aureus* ATCC43300 (CC30/ST39-MRSA-II) at 5 × 10^5^ CFU/mL was incubated with a range of concentrations of vancomycin, rifampicin, vancomycin + rifampicin and PM-399, in 2-fold and 10-fold dilutions. The range of concentrations used was 8 µg/mL to 0.005 µg/mL for vancomycin, 0.128 µg/mL to 0.001 µg/mL for rifampicin and 8 + 0.125 µg/mL to 0.005 + 0.001 µg/mL for vancomycin + rifampicin, respectively. PM-399 concentration ranged from 1.6 × 10^4^ PFU/mL to 6.25 × 10^−2^ PFU/mL. MICs were determined following the CLSI standard for aerobic bacteria by the microbroth dilution method [71] with the deviation that BHI medium was used instead of MHB.

Bacteria were collected from wells treated with the dilution below MIC to start the consecutive round. This procedure was repeated for 21 rounds. The passaging of rifampicin was stopped after development of full resistance. The analysis was done in triplicates. MIC fold change was calculated as the ratio between the MIC measured in each round and the initial MIC.

The rate of spontaneous resistance formation against PM-399 was assessed in nutrient broth [63] *S. aureus* ATCC43300 (CC30/ST39-MRSA-II) grown in 1000 mL of BHI broth to a total of 10^11^ CFU (i.e., 10^8^ CFU/mL or OD600~0.2) was infected with 10^12^ PFU of phage (cocktail PM-399 or individual components, PM4, or PM93). After overnight incubation at 37 °C, the lysate was plated onto Tryptone Soya Agar (TSA, BD, Eysins, Switzerland) plates, then incubated for 24 h at 37 °C. Bacteria were re-streaked into TSA twice to avoid phage presence. A representative number of bacterial colonies (10) were then tested for phage susceptibility, as described in Section 4.4. The resistance rate was calculated as the ratio between the number of bacteriophage-resistant colonies and the initial number of colonies. The experiments were conducted in triplicate for PM-399 and in duplicate for the single phages.

Resistance rate of P66 was measured in nutrient broth. In a 96-well microtiter plate, serial dilutions (1:10) of 180 µL bacteria starting from 10^9^ CFU, were infected with 20 µL of phage, at a constant concentration of 10^8^ PFU. The plate was incubated for 24 h at 37 °C. Survival colonies were tested for phage susceptibility, as described above. The resistance rate was calculated as the ratio between the number of bacteriophage-resistant colonies and the initial number of colonies.

### 4.8. Virulence Index

Phage virulence was measured as described in [5]. Briefly, *S. aureus* strains CC239-MRSA-III (2017-046), CC30-MRSA-IV (2011-278), CC25-MSSA (B91), were grown to log phase, the concentration of bacteria was adjusted and mixed with phage at MOIs of 1, 10, 100, 1000, 10,000 and 100,000. Phages PM4, 2002, 812 and BT3, were used at 5 × 10^8^ PFU/mL. Reaction plates were incubated at 37 °C for 24 h. OD600 was measured every 5 min and further analyzed in Excel.

The area under the OD600 curves was calculated from time of infection until 24 h, for the phage and for the buffer-treated samples. The blank area (background OD) was subtracted in both cases. The local virulence of each phage at each MOI was calculated as the ratio of the area under the OD600 curve of the phage-treated and untreated samples, as depicted in Figure 3A.

To analyze the resistance status of the bacteria after phage treatment, liquid from the well corresponding to the lowest MOI at which bacterial outgrowth was seen was plated on TSA. Bacteria were re-streaked onto TSA twice to remove phages. The EOP of the test phages/cocktail was measured on three colonies from each well and the respective wild type strain as described under “plaquing host range”.

### 4.9. Genomic Analysis

DNA was extracted from phage lysates by using either a specific phage DNA isolation kit (Norgen Biotek Kit, Norgen Biotek Corp., Thorold, ON, Canada) or by using the DNeasy Blood and Tissue Kit (QIAGEN, Hilden, Germany) as reported by [72]. In some instances, phages were concentrated by precipitation with 10% PEG 8000 and 1M NaCl prior DNA extraction as described by [73]. Sequencing libraries were prepared with a TruSeq kit or tagmentation kit (FC-121-1030, Illumina, San Diego, CA, USA), in case of phages 812, 2002, BT3. The samples were sequenced on Illumina MiSeq v3 2 × 300 bp, and the reads were assembled with Spades [74].

Long terminal repeats in the genomes can lead to assembly artifacts [75]. To solve this issue, some phages (Remus, Romulus,812, BT3, 2002, PM4, PM9, PM32, PM56 and PM93) were re-sequenced by NanoPore. The Sequencing library was prepared with the Rapid Sequencing Kit (SQK-RBK004, Oxford Nanopore Technologies, Oxford, UK) and sequenced with a MinION flow cell, on a MinION device. Nanopore reads were assembled by using canu-1.9, and Illumina reads were used to correct the Nanopore sequencing errors. The designation of genome modules was done according to literature data [25,41] and annotations.

We found discrepancies in the genomes of 812 obtained from the d’Herelle collection and the published sequence MH844528.1. Therefore, we submitted the sequence determined for the version of 812 analyzed in this study to NCBI under accession number (MW546072). The genomes of the Remus and Romulus versions obtained from the d’Herelle collection were identical to the published sequences (JX846612.1 and JX846613.1, respectively), but we submitted new versions complemented with LTRs assembled with combined Illumina and OxNanopore data as described above.

The lytic lifecycle of the ε^2^-phages was confirmed computationally using PHACTs [76]. Antibiotic-resistance genes were analyzed by uploading the translated sequences to ResFinder 4.1 [77], using parameters for *S. aureus* species, targeting chromosomal point mutations and acquired antimicrobial resistance genes. Absence of virulence factors was analyzed by BLASTn, against the Comprehensive Antibiotic Resistance Database (CARD, v3.1.0 of 6 August 2020) [78]. Absence of virulence genes was further confirmed with was done by Virulence Finder 2.0 CGE, [79] against *S. aureus*.

Taxonomic classification was inferred from closest sequenced relatives identified by BLAST analysis of phage sequences to NCBI’s RefSeq database.

## 5. Patents

Two patent applications with method claims and composition of matter claims related to this study were filed at the European Patent Office (Application numbers EP20185697.8 and EP20185700.0).

## Figures and Tables

**Figure 1 pharmaceuticals-14-00325-f001:**
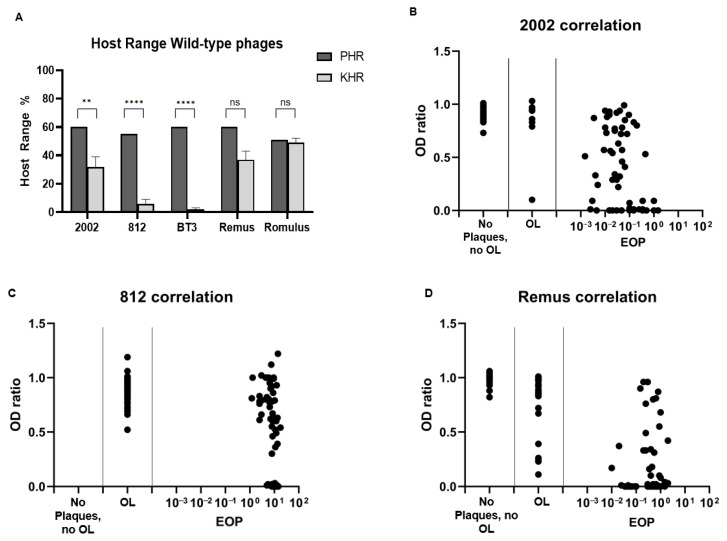
Plaquing and kinetic host range of wild-type phages. (**A**) Comparison of plaquing and kinetic host range over the panel of 110 strains for the wild-type phages. Kinetic host range is lower than plaquing host range for all phages tested. The statistics was calculated by one-way ANOVA for all groups against all groups. **** *p* < 0,0001; ** *p* < 0,0021; ns, not significant. (**B**–**D**) correlation of OD ratio and EOP for phages 2002, 812 and Remus. The OD ratio represents the ratio of OD 600 after 24 h of phage treated vs. control-treated suspensions; an OD ratio value of 1 indicates that phage-treated bacteria grew to the same OD as the growth control, a value of 0 indicates that bacterial growth was fully suppressed. Opaque lysis (OL) is defined as absence of plaque formation, while at the same time the phage lysate inhibits bacterial growth on a double agar layer plate where it is spotted.

**Figure 2 pharmaceuticals-14-00325-f002:**
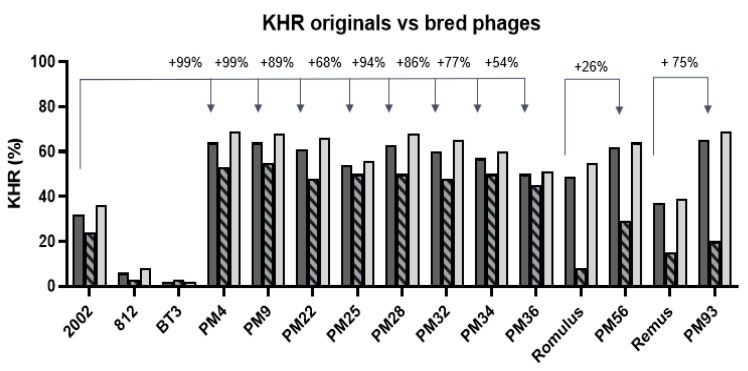
Improvement of kinetic host range of ε^2^ phages. Kinetic host ranges of the bred bacteriophages and their ancestors calculated for the overall panel of 110 *S. aureus* strains (dark grey bars). The data representing the overall KHR of the ancestor phages is identical to the KHR data in Figure 1A. For the breeding strains (dashed bars) and the strains not used for breeding (light grey bars). The KHR was improved on strains used for breeding and on strains not used for breeding. Improvement is indicated as a percentage calculated over the wild type ancestor with the highest KHR value, as indicated with the arrows, from unrounded KHR values.

**Figure 3 pharmaceuticals-14-00325-f003:**
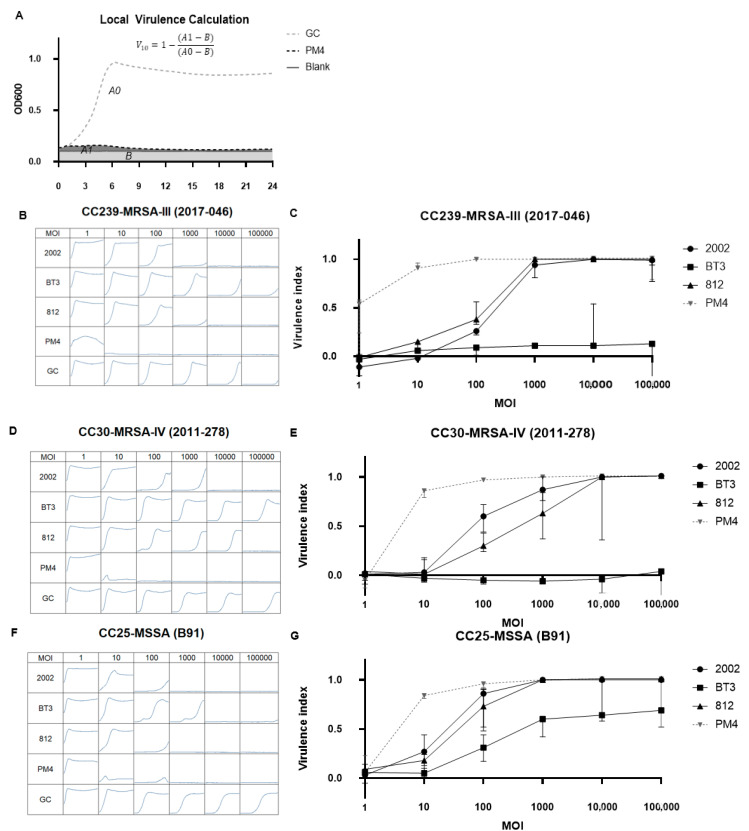
Comparison of the virulence of PM4 and its ancestors 2002, BT3 and 812. (**A**) Illustration of the calculation of local virulence V_10_. Phage virulence at a defined MOI is termed local virulence V_10_ (the subscript indicates the MOI), calculated with the formula on top of the graph. A1 and A0 are the areas under the OD600 curves of the treated and untreated samples, respectively. B is the blank area. (**B**,**D**,**F**) depict the lytic activity of the bred bacteriophage PM4 and its ancestors at different starting MOIs and on three different strains: CC239-MRSA-III (2017-046), CC30-MRSA-IV (2011-278), CC25-MSSA (B91), respectively. Graphs show optical density measurements at 600 nm (OD600) of bacterial suspensions in presence or absence (GC: growth control) of the bacteriophages in a 96-well plate for 24 h at 37 °C. The bacterial strain name is depicted as title, the bacteriophages on the left and the different MOIs at the top of each column. The concentration of the phages was kept constant at 5 × 10^8^ PFU/mL and starting CFU/mL were modified to reach the different MOI concentrations. (**C**,**E**,**G**), depict virulence curves of the three ancestor phages and PM4. The virulence curves were calculated by plotting the local virulence as a function the starting MOI used in the experiment.

**Figure 4 pharmaceuticals-14-00325-f004:**
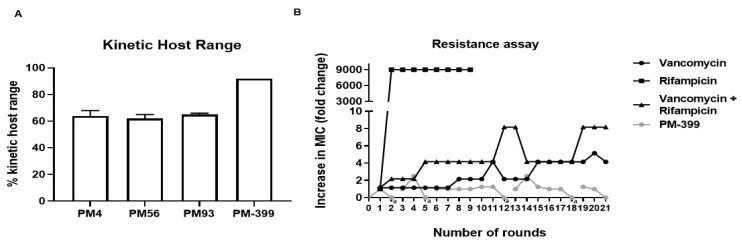
(**A**) Kinetic host range of the cocktail PM-399 and the 3 constituent ε^2^-phages PM4, PM93 and PM56 relative to the panel of 110 *S. aureus* strains. Two repetitions were measured, of which the average and standard deviation is shown. (**B**) Increase in Minimum Inhibitory Concentration (MIC) over 21 rounds of passaging for PM-399, vancomycin, rifampicin and the mix of vancomycin + rifampicin on *S. aureus* CC30-ST39-MRSA-II-ATCC43300. (†): In some passaging rounds, an MIC of 0 was measured for the PM-399 cocktail (i.e., no bacterial growth even without adding phage), which indicates that phages from the previous round of passaging were carried over. In these cases, the experiment was re-started with wild-type bacteria.

**Figure 5 pharmaceuticals-14-00325-f005:**
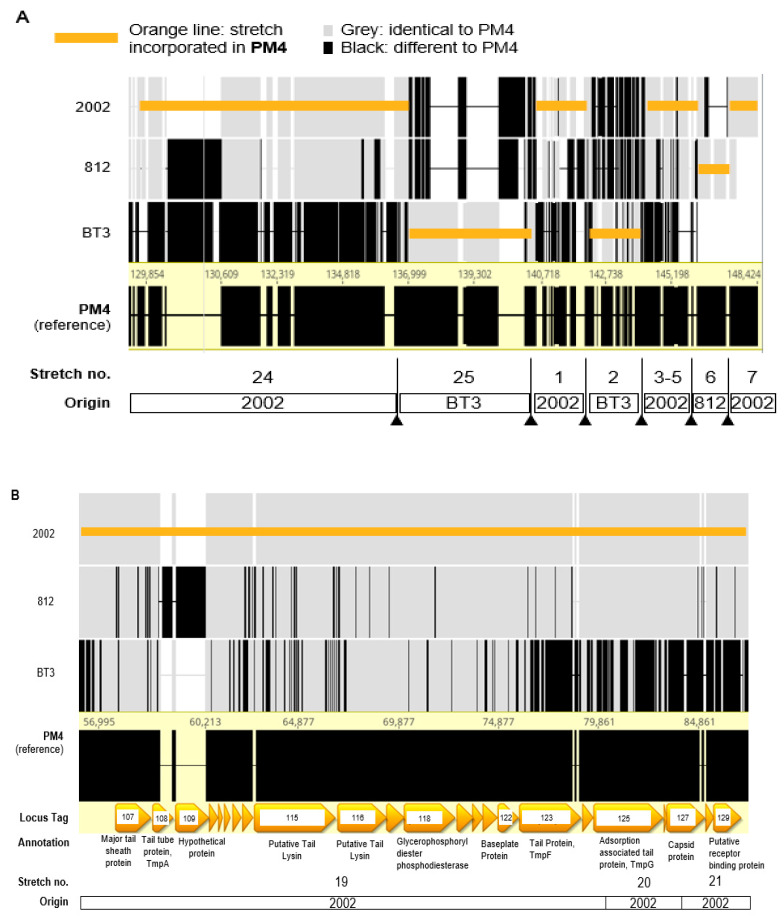
PM4 genome intercrossing in different stretches. Graphical overview of some of the stretches represented in Table 3, for phage PM4 and its homology with the ancestor phages 2002, 812 and BT3. The genome of PM4 is a mosaic of genetic intercrossing between ancestor phages 812, BT3 and 812. Grey areas in the ancestor genomes are identical to PM4, black stretches or vertical lines indicate differences. The orange line indicates the origin of each region incorporated in the genome of PM4. Arrow heads indicate recombination sites. (**A**) Representation of the stretches 24, 25 and 1 to 7 (LTRs), from bases 129,854 to 148,424. (**B**) Overview of stretches 19 to 21, from bases 56,995 to 87,000, located in the structural module, where tail fiber genes and RBPs are clustered. Annotation of some genes is indicated, referring to the Locus Tag number. There are no recombinations or any changes vs. 2002. (**C**) Detailed view of the lysis module, from bases 32,165 to 38,582. Annotation of each gene is indicated, referring to the Locus Tag number. Mutation on PM4, not corresponding to any of the ancestors, is indicated as a green square. Dashed orange lines in 812 indicate that the stretch in PM4 is identical to 2002 and 812.

**Table 1 pharmaceuticals-14-00325-t001:** Overview of CCs of *S. aureus* strains used in this study. A comparison between the % of strains by CC in this study and literature range is shown (See Appendix A for further information).

Clonal Complex	# of Isolates Used in This Study	% of Total Strains in This Study	% MRSA	Literature Range % of Strains ^a^
CC45	19	17%	16%	4–26%
CC30	16	15%	31%	4–23%
CC8	15	14%	40%	7–27%
CC22	9	8%	89%	2–3%
CC5	8	7%	88%	8–35%
CC398	7	6%	100%	1%
CC25	6	5%	0%	7–9%
CC101	3	3%	0%	8%
CC12	3	3%	0%	5%
CC15	3	3%	0%	6–14%
CC80	2	2%	100%	1%
CC9	2	2%	0%	2%
CC88	2	2%	0%	2%
CC6	2	2%	100%	N/A
CC239	2	2%	100%	4%
CC1	2	2%	100%	2–8%
CC96	1	1%	0%	1%
CC59	1	1%	100%	1%
CC121	1	1%	0%	3%
CC7	1	1%	0%	3%
CC97	1	1%	0%	1%
CC772	1	1%	100%	N/A
CC60	1	1%	100%	N/A
CC395	1	1%	0%	N/A
CC49	1	1%	0%	2%
Total no.	110	100%	43%	-

^a^ Kanjilal et al., 2018; Arias et al., 2017; Rasmussen et al., 2013, Lüdicke et al., 2010.

**Table 2 pharmaceuticals-14-00325-t002:** Summary of the phages described in this study.

Phage Name	Reference	Genome Length (bp)	Accession Number
2002	This study	145,076	MW528836
BT3	This study	146,878	MW546073
812	Pantůček et al., 1998 This study	150,390 148,660	MH844528.1 MW546072
Remus	Vandersteegen et al., 2013 This study	134,641 141,985	JX846612 MW546076
Romulus	Vandersteegen et al., 2013 This study	131,332 136,651	JX846613.1 MW546077
PM4	This study	148,627	MW546074
PM9	This study	148,495	MW546064
PM22	This study	140,558	MW546065
PM25	This study	142,467	MW546066
PM28	This study	142,406	MW546067
PM32	This study	148,303	MW546070
PM34	This study	143,062	MW546068
PM36	This study	143,035	MW546069
PM56	This study	136,653	MW546071
PM93	This study	144,038	MW546075
P66	Kwan et al., 2005	18,199	NC_007046.1
P68	Vybiral et al., 2003	18,227	NC_004679.1

**Table 3 pharmaceuticals-14-00325-t003:** Genome stretches of ε2-phages bred from BT3, 2002 and 812. The stretch boundaries are given in the first two columns and refer to PM4. The third column represents the genome module, which was divided in LTR, replication and transcription module, structural genes, and lysis module. Genome stretches where the assignment was not clear are indicated as unassigned. Each cell represents a stretch of the indicated phage, the color denotes the homology of given stretch compared to the ancestors: dark grey for 812, grey for BT3, light grey for 2002 and white for regions with recombinations. The two LTRs are composed of 1 to 7, which are therefore repeated at the end of the genomes. The stretches from 8 to 25 involve sections of the genome between the two LTRs.

From (nt)	To (nt)	Module	Stretch	PM4	PM25	PM22	PM32	PM28	PM34	PM36
1	2299	LTR (left)	1	2002	BT3/812	BT3/2002	2002	2002	812	812
2300	2885	LTR (left)	2	BT3	812	BT3	BT3	BT3	BT3	812
2886	4248	LTR (left)	3	BT3/2002	812	2002/BT3	2002/BT3	BT3/2002	BT3/2002	812
4249	5604	LTR (left)	4	2002	812	2002	2002	2002	2002	812
5605	6298	LTR (left)	5	2002/812	2002/812/BT3	2002/812	2002/812	2002/812	812	812
6299	7508	LTR (left)	6	812	812	812	812	812	812	812
7509	8343	LTR (left)	7	812/2002	812	812/2002	812/2002	812/2002	812	812
8344	10,181	Replication	8	2002	2002	2002	2002	2002	2002	2002
10,182	10,849	Replication	9	2002	2002	2002	2002	2002	812	2002
10,850	11,837	Replication	10	2002	2002	2002	2002	2002	2002	2002
11,838	21,147	Replication	11	2002	2002	2002	2002	2002	2002	812/2002
21,148	34,935	Replication Lysis	12	2002	2002	2002	2002	2002	2002	812
34,936	35,732	Lysis	13	BT3	BT3	BT3	BT3	BT3	BT3	BT3
35,733	43,441	Lysis Structural	14	2002	2002	2002	2002	2002	2002	2002
43,442	45,265	Structural	15	2002	2002	2002	2002	812	2002	2002
45,266	46,723	Structural	16	2002	2002	2002	2002	2002	2002	2002
46,724	48,054	Structural	17	2002	2002	2002	2002	2002	2002	BT3/2002
48,055	57,822	Structural	18	2002	2002	2002	2002	2002	2002	812
57,823	79,061	Structural	19	2002	2002/812	2002	812/2002/BT3	2002/812	2002/812	812
79,062	83,269	Structural	20	2002	2002	2002	2002	2002	2002	2002
83,270	95,880	Structural Replication	21	2002	2002	2002	2002	2002	2002	812
95,881	104,924	Replication Unclear	22	2002/BT3	2002/BT3	2002	2002/812/BT3	2002/BT3	2002/812/BT3	812
104,925	106,889	Unclear	23	2002	2002	2002	2002	2002	2002	812
106,890	137,479	Unclear	24	2002	2002/812	2002	2002/812/BT3	2002	2002/812	812
137,480	140,284	Unclear	25	BT3	BT3	BT3	BT3	BT3	812	812
140,285	142,583	LTR (right)	1	2002	BT3/812	BT3/2002	2002	2002	812	812
142,584	143,169	LTR (right)	2	BT3	812	BT3	BT3	BT3	BT3	812
143,170	144,532	LTR (right)	3	BT3/2002	812	2002/BT3	2002/BT3	BT3/2002	BT3/2002	812
144,533	145,888	LTR (right)	4	2002	812	2002	2002	2002	2002	812
145,889	146,582	LTR (right)	5	2002/812	2002/812/BT3	2002/812	2002/812	2002/812	812	812
146,583	147,792	LTR (right)	6	812	812	812	812	812	812	812
147,793	148,627	LTR (right)	7	812/2002	812	812/2002	812/2002	812/2002	812	812

## Data Availability

The data presented in this study are available within the article and supplementary material.

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
