# Peer review of "ε^2^-Phages Are Naturally Bred and Have a Vastly Improved Host Range in *Staphylococcus aureus* over Wild Type Phages"

_pharmaceuticals, 2021, doi:10.3390/ph14040325_

Round 1

Reviewer 1 Report

Pathogenic bacteria rapidly develop resistance against commonly used drugs at an alarming rate, and antibiotic resistance is one of the biggest problems we encounter globally. In recent years phage therapy emerged as an alternative approach to standard treatments to combat bacteria resistant to antibiotics. All topics associated with phage therapy are nicely outlined in the Introduction, containing clearly formulated objectives. However, it would be beneficial to the reader to bring few words about the biology of phages used in the experiments. Some information are provided on page 9.

The present submission focuses on lytic phages that target a common pathogen, Staphylococcus aureus, an antibiotic-resistant, and biofilm-forming bacterium. The focus was given to lytic e2–phages bred from wild type ancestors to produce phage variants superior in the killing of target bacteria. The methodology underlying this kind of breeding is rooted in Alfred Hershey experiments based on homologous recombination performed in the late forties of the 20th century Hershey and Rotman, 1949; Genetics 34:44-71). In my opinion, the Authors succeeded in developing a methodology to breed phages with higher virulence than parental lineages. The hybrid phages show a broad plaquing host range. The critical issue is a resistance rate, which in the case of some phages produced using this approach is below 1 cell in 1011.

What is surprising and discussed by the Authors (page 12, lane 334-343) is the lack of correlation between the efficiency of plaquing and killing of bacteria, as shown in Fig. 1 (panel B, C and D).

Panel A of Fig. 1 shows a comparison of plaquing and kinetic host range over the panel of 110 strains of S. aureus listed in Table 1. It is reasonable to expect that the same values should be presented in Fig. 2 (left part), which illustrates kinetic host range of hybrid phages. However, presented data differ. Please explain this discrepancy.

Discussion is crafted skilfully.The results are put in the context of previous findings.

Minor points:

As cell lysis is central to this study, it would be appropriate to say more about the phage lytic module. In the case of phage 812 (GenBank Acc. No. MH844528.1), it seems that there are three genes coding for lytic enzymes: 812_40 (transglycosylase), 812_69 (lytic transglycosylase), and 812_73 (LysK endolysin with the specificity of N-acetylmuramoyl-L-alanine amidase). Lytic enzymes encoded by phage Remus and Romulus share the same specificity with phage 812 endolysin. However, they are different at the aa sequence level. It would be interesting to validate the virulence of the newly produced phages concerning the lytic module. Please, consider expanding Figure 5 to show the genetic content of hybrid phages (at least PM4, PM56 and PM93), and exchange functional modules between ancestral phages. It seems that everything is at hand (see Table 3).

Effective MOI (multiplicity of infection) vary from 10 (phage P4) to 10,000 (phages 2002 and 812; page 6, lane 192). Is there any rational explanation for this difference? Can Authors discuss the effect of phage adsorption on the infection process? What is a receptor for these phages? The Authors wrote about transient tolerance against the phages (page 6, lane 209). Is it possible to give more detail to explain this phenomenon?

Page 12, line 356-357: Authors wrote: “…wild type phages that were known to be highly homologous…”. Please correct this since homology cannot be quantified. Homology exists when analyzed genes, proteins have a common ancestry. Authors should instead use formulation: high identity or similarity.

I failed to retrieve from the GenBank database the nt sequence of phage 2002 (Acc. No. MW528836).

Author Response

Reviewer 1

  • Pathogenic bacteria rapidly develop resistance against commonly used drugs at an alarming rate, and antibiotic resistance is one of the biggest problems we encounter globally. In recent years phage therapy emerged as an alternative approach to standard treatments to combat bacteria resistant to antibiotics. All topics associated with phage therapy are nicely outlined in the Introduction, containing clearly formulated objectives. However, it would be beneficial to the reader to bring few words about the biology of phages used in the experiments. Some information are provided on page 9.
    • As suggested, we have included some general information on the biology of phages used in the experiments, as well as references for more detailed publications were this can be found. See L87-92.
  • The present submission focuses on lytic phages that target a common pathogen, Staphylococcus aureus, an antibiotic-resistant, and biofilm-forming bacterium. The focus was given to lytic e2–phages bred from wild type ancestors to produce phage variants superior in the killing of target bacteria. The methodology underlying this kind of breeding is rooted in Alfred Hershey experiments based on homologous recombination performed in the late forties of the 20thcentury Hershey and Rotman, 1949; Genetics 34:44-71). In my opinion, the Authors succeeded in developing a methodology to breed phages with higher virulence than parental lineages. The hybrid phages show a broad plaquing host range. The critical issue is a resistance rate, which in the case of some phages produced using this approach is below 1 cell in 1011.
  • What is surprising and discussed by the Authors (page 12, lane 334-343) is the lack of correlation between the efficiency of plaquing and killing of bacteria, as shown in Fig. 1 (panel B, C and D).
  • Panel A of Fig. 1 shows a comparison of plaquing and kinetic host range over the panel of 110 strains of S. aureus listed in Table 1. It is reasonable to expect that the same values should be presented in Fig. 2 (left part), which illustrates kinetic host range of hybrid phages. However, presented data differ. Please explain this discrepancy.
    • Thanks for pointing this out. We may have poorly explained this, so we added further explanations in the caption of figure 2, explaining that values of Fig.1A and “overall KHR” values in Fig. 2 are identical. Please let us know if we overlooked some other detail.
  • Discussion is crafted skilfully. The results are put in the context of previous findings.

Minor points:

  • As cell lysis is central to this study, it would be appropriate to say more about the phage lytic module. In the case of phage 812 (GenBank Acc. No. MH844528.1), it seems that there are three genes coding for lytic enzymes: 812_40 (transglycosylase), 812_69 (lytic transglycosylase), and 812_73 (LysK endolysin with the specificity of N-acetylmuramoyl-L-alanine amidase). Lytic enzymes encoded by phage Remus and Romulus share the same specificity with phage 812 endolysin. However, they are different at the aa sequence level. It would be interesting to validate the virulence of the newly produced phages concerning the lytic module. Please, consider expanding Figure 5 to show the genetic content of hybrid phages (at least PM4, PM56 and PM93), and exchange functional modules between ancestral phages. It seems that everything is at hand (see Table 3).
    • PM93 and PM56 mutations are listed in tables 3 and 4 in supplementary material. There are no mutations in the endolysin, or other genes involved in peptidoglycan lysis. We have added the location in functional cassettes to the tables.
    • PM4: we have extended our analysis and added figures 5B and 5C, depicting a detailed view of the mentioned genes, plus the description in results (See L316-337). We have also discussed these findings in the context of previous findings (L446-464). We have also added the functional cassettes of each stretch in table 3, as a reference for the mutations in the new-added figures.
  • Effective MOI (multiplicity of infection) vary from 10 (phage P4) to 10,000 (phages 2002 and 812; page 6, lane 192).
    1. Is there any rational explanation for this difference?
    2. Can Authors discuss the effect of phage adsorption on the infection process? What is a receptor for these phages?
    3. The Authors wrote about transient tolerance against the phages (page 6, lane 209). Is it possible to give more detail to explain this phenomenon?
  • 1- If a phage is more virulent, less phage particles are needed to lyse more bacteria. This hints at parameters of replication cycle that favor a higher basic reproduction number “R0” (analogous to the epidemiological parameter). This could mean that the adsorption rate is higher (e.g. higher affinity of the receptor binding areas), or the burst size is higher or latency period shorter. However, this was not analyzed in detail. We have included this hypothesis in the discussion, L423-425.
  • 2- The phage receptor was not studied here. For other aureus Herelleviridae phages it was found to be the WTA backbone (proved for 812 and K) (Xia et al, 2011). However, phage adsorption is a multistep process – based on our data we cannot narrow it down. In any case adsorption is only one of the factors influencing host range and resistance. We show that there were no mutations in the tail fiber proteins in PM4 compared to the source of that stretch, 2002. Also, PM4 has a strongly increased plaquing host range compared to 2002. This indicates that the increase in host range is mediated either by other genes (e.g. CRISPR targets/anti-CRISPRs, targets of restriction enzymes, host takeover, capsid assembly), and/or it could be mediated by interactions between 2002’s tail fiber genes with the new genes from BT3 and 812. See L446-464.
  • 3- The phenomenon of transient tolerance has recently been described for subtilis phages, as a temporary immunity mechanism, enabling bacteria to block infection by remodeling WTA polymers. This modification reduces phage binding and restricts phage spread. (Tzipilevich et al, 2021, BiorXiv preprint), see L230.
  • Page 12, line 356-357: Authors wrote: “…wild type phages that were known to be highly homologous…”. Please correct this since homology cannot be quantified. Homology exists when analyzed genes, proteins have a common ancestry. Authors should instead use formulation: high identity or similarity.
    • Thanks for the comment, we have updated accordingly, see L418.
  • I failed to retrieve from the GenBank database the nt sequence of phage 2002 (Acc. No. MW528836).
    • The submitted sequences are not yet released, they will be released only upon acceptance of the manuscript. The sequence of phage 2002 is now included in a supplementary file for the reviewers, together with all other phage sequences.

Reviewer 2 Report

The authors herein present a detailed, thorough, interesting study on host range expansion of phages of Staphylococcus aureus. The manuscript is well written and the methodology is well-designed, with results well-described. Of note, the authors took care to highlight only prevalent strains of S. aureus which is to be commended. The paper is impactful and highlights the potential for “directed evolution” of phages, particularly in the clinical sector. I only have minor comments.

Table 1. What does “literature average” mean in this context?

Table 1, footnotes: What does this footnote refer to?

L129: What was the KHR of BT3? This should be the lower end of the range given here.

L130: Do the authors think this result is significant? There was no mention of statistical methods employed in the PHR/KHR assays. Why is this?

Figure 1B: Acronym “OL” should be defined in the script.

L137: What is “opaque lysis”? This word is an oxymoron.

L140: How do the authors define “high EOP”? This is not the case with Figure 1B, many of the EOP values are below 1.

L152: Not all of the wild type phages had a high EOP

L153: How did the authors choose the phages for the cocktail?

L374: It is interesting how the tail fiber regions were not compared between the bred vs. ancestor phages, as these are the key recognition elements and usually determine host range.

L384-385: Authors should expand on this idea.

Figure 1. It is very difficult to legibly make out the names and dates. Greater resolution needed.

Author Response

Reviewer 2

  • The authors herein present a detailed, thorough, interesting study on host range expansion of phages of Staphylococcus aureus. The manuscript is well written and the methodology is well-designed, with results well-described. Of note, the authors took care to highlight only prevalent strains of S. aureus which is to be commended. The paper is impactful and highlights the potential for “directed evolution” of phages, particularly in the clinical sector. I only have minor comments.
  • Table 1. What does “literature average” mean in this context?
    • Thanks for spotting it, it has been updated to “literature range” (see L116).
  • Table 1, footnotes: What does this footnote refer to?
    • The footnote was removed by mistake, it was added back. The footnote refers to the “literature range of strains” (last column in the top row of the table).
  • L129: What was the KHR of BT3? This should be the lower end of the range given here.
    • KHR of BT3 was 2%, we have updated the values accordingly across the manuscript (See L 135).
  • L130: Do the authors think this result is significant? There was no mention of statistical methods employed in the PHR/KHR assays. Why is this?
    • We have added a statistical analysis of the differences between PHR and KHR to Fig. 1A. We also added a statistical analysis to Supplementary Fig. 2, on the difference in KHR between Ɛ2-phages and their ancestors Thanks for the comment, we believe this analysis adds value to the manuscript.
  • Figure 1B: Acronym “OL” should be defined in the script.
    • “OL” is defined:
      • Please see figure 1 caption: Opaque lysis (OL) is defined as absence of plaque formation, while at the same time the phage lysate inhibits bacterial growth on a double agar layer plate where it is spotted.(L158)
      • For clarity we also included this definition in Materials and Methods section 4.4: strains with no plaque formation but only impairment of growth at the site of the phage spot (“opaque lysis”) (L509)
      • For lack of a better expression, we are using this term, which has been used by other authors to describe the phenomenon we are defining (Caflisch & Patel, 2019; Merabishvili et al., 2009)
    • L137: What is “opaque lysis”? This word is an oxymoron.
      • See above.
    • L140: How do the authors define “high EOP”? This is not the case with Figure 1B, many of the EOP values are below 1.
      • We have updated the text to “EOP close to 1” (L146) to avoid confusion.
    • L152: Not all of the wild type phages had a high EOP
      • We have updated the text to “forming plaques” to avoid confusion in the text. See L162.
    • L153: How did the authors choose the phages for the cocktail?.
      • The phages 812, 2002 and BT3 belong to the Kayvirus genus, they were mixed and bred as a cocktail, since recombination upon superinfection was expected to occur due to their nucleotide similarity. Phages Remus and Romulus, belonging to the Silviavirus genus, were not expected to recombine with Kayvirus, thus were bred individually. We have updated the text to introduce this, See L163-165.
    • L374: It is interesting how the tail fiber regions were not compared between the bred vs. ancestor phages, as these are the key recognition elements and usually determine host range.
      • Thanks to your comment, we have updated figure 5 were we now show more detailed analysis of the lysis cassette and the genes encoding for tail fibers. We´ve discussed it in the text, according to previous findings. We believe this comment was very constructive. See L316-337 in results and L446-462 in discussion.
    • L384-385: Authors should expand on this idea.
      • We have expanded the analysis considering previous findings. See L458-464.

    • Figure 1. It is very difficult to legibly make out the names and dates. Greater resolution needed.
      • We have now updated it and added in greater resolution. We have also attached it as a separate file, together with the supplementary material in the submission.